# The Value of Repeat 5-HIAA Measurements as a Predictor of Carcinoid Heart Disease: A Prospective 5-Year Follow-Up Study in Patients with Small Intestinal Neuroendocrine Tumors

**DOI:** 10.3390/cancers16233896

**Published:** 2024-11-21

**Authors:** Iiro Kostiainen, Piia Simonen, Katri Aaltonen, Riikka Lindén, Noora Karppinen, Daniel Gordin, Janne Rapola, Camilla Schalin-Jäntti, Niina Matikainen

**Affiliations:** 1Endocrinology, Abdominal Center, Helsinki University Hospital and University of Helsinki, ENDO-ERN (European Reference Network on Rare Endocrine Conditions), 00280 Helsinki, Finland; iiro.kostiainen@hus.fi (I.K.); niina.matikainen@hus.fi (N.M.); 2Cardiology, Heart and Lung Center, Helsinki University Hospital and University of Helsinki, 00280 Helsinki, Finland; 3Radiology, HUS Diagnostic Center, Helsinki University Hospital and University of Helsinki, ENDO-ERN (European Reference Network on Rare Endocrine Conditions), 00260 Helsinki, Finland; riikka.linden@hus.fi; 4Department of Nephrology, Abdominal Center, Helsinki University Hospital and University of Helsinki, 00280 Helsinki, Finland; 5Minerva Institute for Medical Research, 00290 Helsinki, Finland; 6Joslin Diabetes Center, Harvard Medical School, Boston, MA 02115, USA

**Keywords:** small intestinal neuroendocrine tumor, carcinoid heart disease, proBNP, 5-HIAA, cumulative exposure

## Abstract

Small intestinal neuroendocrine tumors (SI-NETs) can lead to carcinoid syndrome and carcinoid heart disease (CHD). In this prospective study we aimed to identify early risk markers for CHD and mortality in patients with SI-NETs. We measured basal serum 5-HIAA and cumulative 5-HIAA (Cum-5-HIAA) based on repeated measurements, proBNP, vascular function, transesophageal echocardiography (TTE), and hepatic tumor load in 65 patients with SI-NETs during the median follow-up of 5 years in 54 of the patients who underwent prospective follow-up. Survival was evaluated during the median follow-up of 6 years. Three patients had CHD at baseline and two (4%) developed CHD. Cum-5-HIAA and proBNP correlated with CHD. Of note, Cum-5-HIAA was the best biomarker for CHD, outperforming pro-BNP, chromogranin A (CgA), and individual 5-HIAA. The strongest predictors of mortality were the diagnosis of CHD and high liver tumor burden, while high blood vessel stiffness was also associated with lower survival rates. The incidence of CHD was low during follow-up, probably reflecting efficient treatment regimens. These findings may help identify patients at high risk for CHD and mortality and guide early interventions to improve patient outcomes.

## 1. Introduction

Small intestinal neuroendocrine tumors (SI-NETs) are rare malignancies that originate from the enterochromaffin cells of the small intestine [1]. SI-NETs frequently excrete serotonin and other vasoactive agents, which can lead to the development of carcinoid syndrome (CS) and its hallmark symptoms of chronic diarrhea and flushing [2,3]. While it has been confirmed that serotonin secretion plays a role in the development of CS, the significance of other humoral mediators is still under investigation [3].

Carcinoid heart disease (CHD) is a rare and serious complication thought to be caused by exposure to non-physiological serotonin levels [3] and involves predominantly right-sided cardiac abnormalities and increased mortality and morbidity [4,5,6]. CHD is characterized by the fibrosis of the cardiac valves, particularly the tricuspid and pulmonic valves, with subsequent regurgitation, stenosis, and heart failure [2]. This fibrosis is thought to be mediated by the excessive stimulation of the 5-hydroxytryptamine-2B receptor on cardiac fibroblasts, which induces their activation [3,7]. Furthermore, CHD-like disease is evident in different animal models and after exposure to serotoninergic drugs [8,9,10]. In human studies, only indirect evidence of serotonin’s role has been obtained: elevated levels of the serotonin metabolite 5-hydroxyindoleacetic acid (5-HIAA) have been associated with CHD diagnosis and progression [11,12,13,14].

In terms of the clinical diagnosis of CHD, several scoring systems are utilized to evaluate cardiac echocardiography results [2,15,16]. However, the tools available for determining the prognosis of CHD remain suboptimal. Current guidelines recommend screening individuals with symptomatic CS or evidence of elevated serotonin exposure for CHD, and N-terminal pro-brain natriuretic peptide (proBNP) is considered the most useful biomarker for identifying CHD in CS patients [2]. The recent European Neuroendocrine Tumor Society (ENETS) guidelines suggest that a proBNP concentration of <260 ng/L has a negative predictive value of 97% and, accordingly, that those with a proBNP concentration of >260 ng/L should be referred for echocardiography [2]. However, a high proBNP level is not specific to CHD, and some studies have reported divergent results. For example, a recent study found that proBNP was a poor prognostic indicator and that plasma 5-HIAA levels could be used to differentiate patients with CHD [17].

In recent years, the treatments for SI-NETs have evolved. Most importantly, there have been promising developments in peptide receptor radionuclide therapy (PRRT). However, metastasis and CS occur frequently in patients with SI-NETs, [2] and the prognostic factors that affect their survival remain unclear. In addition, the ENETS guidelines on CS and CHD [2] outline several unmet needs in the field of diagnostics and prognostics. These include the need for further implementation of blood-based assays for serotonin and 5-HIAA and the evaluation of the prognosis of patients with SI-NETs and CS.

Therefore, in this study, we aimed to address unmet needs in the areas of SI-NET prognosis and CHD diagnosis by conducting a prospective follow-up of SI-NET patients who had undergone transthoracic echocardiography (TTE) at baseline and approximately five years later. We utilized a diverse range of baseline and follow-up data that were derived from vascular function measurements, TTE scores, quantitative hepatic tumor burden assessments, and serial biomarker analyses to predict CHD development and mortality among SI-NET patients.

## 2. Patients and Methods

### 2.1. Patients and Data

The study population consisted of patients with a histologically confirmed diagnosis of SI-NET (Orpha-codes 423975 and 100093) who were treated at the Finnish Reference Center for Rare Endocrine Conditions (EndoERN FIN) at Helsinki University Hospital between May 2016 and November 2017. All participants gave their written informed consent. Subjects with hereditary tumor predisposition syndromes were excluded. The study was conducted in accordance with the Declaration of Helsinki and approved by the Ethics Committee of Helsinki University Hospital (30/13/03/01/16). The patients’ baseline characteristics, including their vascular function measurements, are described in more detail in our previous study [18].

All available patients were invited to take part in a prespecified follow-up, which included the collection of TTE and plasma biomarker data. Follow-up measurements were performed between June 2021 and April 2023. Figure 1 depicts the process of acquiring the follow-up TTE data. All but three of the patients that could be contacted were recruited for the follow-up portion. One of these patients had other follow-up TTE data that could be used in the study. Among the 17 patients who died before the initiation of the follow-up examinations, nine had other available follow-up TTE data that could be utilized in the analyses. The median time from the initial assessment to follow-up TTE was 61 months.

### 2.2. Clinical Characteristics

Other clinical data and laboratory results were collected from the patients’ electronic records. Hepatic tumor burden was evaluated by an abdominal radiologist who referred to clinical images captured within a year from the follow-up TTE. The protocol used to assess the hepatic tumor burden is described in our previous work [18], with metastatic tumor burden split into categories of 0%, 1–10%, 11–25%, 25–50%, and over 50% of the liver. Serum 5-HIAA (S-5-HIAA), proBNP, and most chromogranin A (CgA) analyses were performed at the Helsinki University Hospital laboratory. S-5-HIAA was measured using liquid chromatography–mass spectrometry [19] with the upper limit of normal (ULN) set to 123 nmol/L, proBNP was measured using an immunochemiluminometric assay, and CgA was measured using either a radioimmunoassay or time-resolved amplified cryptate emission. Some of the CgA analyses were outsourced to accredited laboratories. As the reference range for CgA varied throughout the study period, the values were scaled to a ULN.

Cumulative exposure to S-5-HIAA exceeding the upper limit of normal (Cum-5-HIAA) was calculated using all available S-5-HIAA measurements. The latest available TTE data were selected for the analysis of Cum-5-HIAA, unless CHD was diagnosed from the baseline TTE, in which case the baseline TTE data were used. The median number of S-5-HIAA measurements available for analysis was 15 (interquartile range (IQR) 10–19), taken over a median period of 86 months (IQR 60–101). Cum-5-HIAA was calculated using the following formula:$$\left(\sum_{i=m}^{n-1}E_{i}\times \left(t_{i+1}-t_{i}\right)\right)+E_{n}\left(t_{TTE}-t_{n}\right)$$
where *E* is the S-5-HIAA exceeding the ULN divided by the ULN; *t* is the time at which the S-5-HIAA was measured; *m* and *n* are the initial and final S-5-HIAA measurements, respectively; and *t_TTE_* is the timing of the TTE after the last S-5-HIAA measurement. Time was measured in years to express the cumulative exposure in ULN years.

### 2.3. Survival

To evaluate overall survival, data on the patients’ vital status and cause of death were collected from their electronic records on 21 September 2023. In the survival analysis, the median duration of the follow-up period was 72 months (IQR 45–82).

### 2.4. Echocardiography

The TTE protocol utilized in this study is described in detail in our previous publication [18]. The diagnosis of CHD was based on echocardiographic grading of tricuspid regurgitation, leaflet mobility, and morphological abnormalities of the leaflets. From these parameters, we calculated the Westberg score, which has a good capacity to discriminate between individuals with and without CHD [20]. The high feasibility of calculating the Westberg score [15] was important in this study because some of the utilized follow-up TTE examinations were not performed with the study protocol. All the included TTE examinations carried out outside of the study were retrieved from digital patient records and re-evaluated. There were seven follow-up TTE examinations (13%) that did not enable complete scoring with the Westberg criteria. None of the patients with incomplete scores were considered to have CHD. A score of three was set as the cutoff for CHD, with previous work showing that high sensitivity and specificity were achieved when this cutoff score was used [15].

### 2.5. Statistical Analysis

Statistical analyses were performed using R statistical software (v4.2.1; R Core Team 2022). Data are presented as medians and interquartile ranges for continuous variables. The categorical variables are presented as frequencies and proportions. Due to the non-normal distribution of biomarkers, nonparametric tests were used for the statistical evaluation of continuous variables, with correlations assessed using Spearman’s correlation coefficient, group differences for continuous variables assessed using the Mann–Whitney U test, and group differences for categorical variables assessed using Fisher’s exact test.

Time-to-event (mortality) analyses were performed using Kaplan–Meier estimates, log-rank tests, and a Cox proportional hazards model. For the Cox proportional hazard model, variables were tested for collinearity, and Schoenfeld residuals were calculated to confirm that the proportional hazard assumption was not violated. No significant collinearity or violation of the proportional hazard assumption was noted. The Cox proportional hazards model was fitted with sex, age, presence of CHD at initial assessment, presence of liver metastases, S-5-HIAA, and aortic PWV data.

For the receiver operator characteristic (ROC) analysis, models were built with the assumption that higher biomarker concentration or exposure was associated with CHD. One patient was excluded from the Cum-5-HIAA ROC analysis as they had already developed CHD at the time of the initial SI-NET diagnosis. An area under the curve (AUC) ≥ 0.80 in the ROC analysis was considered to be indicative of a good diagnostic performance [21]. The reported *p*-values are two-sided, with a *p*-value < 0.05 considered statistically significant. In case of incomplete data for the statistical test, data were omitted from the analysis.

## 3. Results

### 3.1. Patient Characteristics

The patients’ characteristics, hepatic tumor burden, biomarker measurements, and the treatments they received, at both baseline and during the follow-up period, are shown in Table 1. Regarding the subset of patients who developed CHD, a separate column of data is provided that shows the variable data obtained at the time of the CHD diagnosis. Treatments received at different time points prior to assessment are shown in Table 1.

### 3.2. Echocardiographic Features Observed During the Follow-Up

Three of the patients had CHD at baseline. Among the patients who were not diagnosed with CHD based on the initial TTE results, two patients (3.7%) were diagnosed with CHD during the median 61-month follow-up.

The patients’ echocardiographic features at baseline and follow-up are shown in Table 2. No statistically significant changes from the baseline measurements were noted for any variable. Minor fluctuations in measured regurgitation were frequent, with either decreases or increases in regurgitation classes noted in 29% of tricuspid valves and 30% of pulmonic valves.

### 3.3. Correlations Between Biomarkers and Westberg Score

The correlations between the potential biomarkers and the Westberg score were assessed. Significant correlations were noted between the Westberg score and proBNP (Spearman’s ρ = 0.31, *p* = 0.02) and Cum-5-HIAA (Spearman’s ρ = 0.32, *p* = 0.01). Scatter plots and Spearman correlation values for these biomarkers and the Westberg score are shown in Figure 2.

### 3.4. Assessment of Patients Who Did Not Undergo Follow-Up TTE

Patients who did not undergo follow-up TTE were further evaluated, and their baseline characteristics are shown in Table 1. Compared to the patients who underwent follow-up TTE, those who did not undergo follow-up TTE did not statistically significantly differ in their age, sex, primary tumor Ki-67, hepatic tumor load, or biomarker measurements at baseline; however, their aortic pulse wave velocity (PWV) values were higher (*p* = 0.03). The median Cum-5-HIAA at death or the end of the follow-up was 3.5 ULN years (IQR 0–25, range 0–35).

### 3.5. Biomarker Performance of the Assessed Variables for the Detection of CHD

The biomarker performance of individual proBNP, CgA, and S-5-HIAA measurements and Cum-5-HIAA for the detection of CHD was assessed via ROC analysis. This analysis included data from patients who were diagnosed with CHD either at the initial assessment or follow-up. The capacity of the different variables to act as biomarkers for CHD is shown in Figure 3. Cum-5-HIAA had the largest AUC at 0.98 (95% confidence interval [CI] = 0.94–1.00) and superior performance, reaching statistical significance and surpassing the preset AUC threshold of 0.80 for good diagnostic performance. The optimal cutoff point was 34.7 ULN years with 100% sensitivity and 95% specificity.

We tested the performance of proBNP using a cutoff value of 260 ng/L, which is suggested in the ENETS guidelines [2] for selecting patients for further CHD screening. In our patient cohort, the negative predictive value (NPV) was 95%, while the positive predictive value (PPV) was only 21%. The performance of proBNP was better at higher cutoff values, with both the NPV and PPV returning higher values at 500 ng/L (96% and 33%, respectively) and 1000 ng/L (96% and 60%, respectively).

### 3.6. Survival and Predictors of Mortality in SI-NET Patients

By the end of the follow-up (median 72 months), 22 (34%) patients had died. The vital status of one patient could not be assessed as they had moved abroad. Median survival was not reached in the Kaplan–Meier estimates at the maximum follow-up of 90 months. In 17 cases (89%), death was due to SI-NET. In two cases (11%), death was not related to SI-NET (glioblastoma and septic cholecystitis). The cause of death in three patients could not be determined due to insufficient data.

Patients who died during the follow-up period had higher baseline hepatic tumor burden, CgA, S-5-HIAA, and Cum-5-HIAA levels and had received interferon therapy more often. Table 3 shows the patient characteristics at baseline by vital status at the end of the follow-up period. High baseline hepatic tumor load, high S-5-HIAA, CHD diagnosis, and high aortic PWV were significantly associated with reduced survival in a single variable assessment performed using a log-rank test. The Kaplan–Meier estimates and log-rank test results for survival are shown in Figure 4.

In the multivariate Cox proportional hazards model, the CHD diagnosis, presence of liver metastases, and high aortic PWV were statistically significantly associated with increased mortality (hazard ratios of 36.1 [95% CI = 5.36–243], 5.28 [1.09–25.6], and 1.22 [1.04–1.43], respectively). Table 4 displays hazard ratios from univariate and multivariate Cox proportional hazard models.

## 4. Discussion

In this prospective study, TTE and biomarker data were collected from well-characterized SI-NET patients over five years. The data analysis showed that the observed incidence of CHD during follow-up was 3.7%, which is substantially lower than that reported in previous studies [22,23,24]. The results are encouraging and may be best explained by the timely and efficient sequencing of treatment regimens in the patients with SI-NETs treated at our European Reference Center for Rare Endocrine Conditions. The main finding of this study was that cumulative 5-HIAA emerged as a novel biomarker and exhibited good performance, surpassing other previously known and potential biomarkers of CHD in ROC analysis. Less severe valve regurgitations were frequently observed, indicating the limited prognostic potential of serial TTE in the prediction of CHD. However, despite the patients receiving modern treatments, which frequently included PRRT, and low CHD incidence, the mortality rate still reached 34% during the 72-month median follow-up.

Given that the development of CHD is attributed to the excessive serotonin produced by NETs [2] and that other studies have noted higher baseline 5-HIAA levels to be associated with an increased risk for the development of CHD [13,22,24], it is understandable that quantifying cumulative serotonin exposure could enable more precise prediction and diagnostics of CHD, as seen in our study. It is difficult to directly measure circulating serotonin; therefore, the levels of its metabolite 5-HIAA in either urine or serum/plasma are often measured [2,19]. To our knowledge, no other study has assessed the performance of cumulative 5-HIAA exposure in CHD diagnostics.

Although our choice of the equation for calculating the cumulative S-5-HIAA exposure likely led to an underestimation of the true exposure in progressive disease, we selected this simple equation as we considered other alternatives to be less precise and practical. For example, calculating and using the average S-5-HIAA level from the start to the end of the follow-up period would have led to an overestimation of the exposure in cases where disease progression was rapid, and S-5-HIAA follows an exponential curve. Similarly, using average values would have led to an underestimation of the exposure after surgery or other treatments that significantly reduced the 5-HIAA burden. Simple equations, such as the one used in this study, are more accessible to clinicians and thus more likely to be used in clinical settings. It should also be noted that if the 5-HIAA level is substantially elevated at the initial presentation, then the measured cumulative exposure might represent only a minor fraction of the true exposure to date, and any diagnostic cutoff values would be unreliable. The findings of this study indicate that cumulative S-5-HIAA exposure is a promising biomarker for use in the long-term surveillance of patients with limited tumor burden at the initial assessment. This constitutes a substantial group of patients, as the incidental discovery of less-advanced SI-NETs is becoming more frequent [25,26].

In this study, the incidence of CHD was low compared to that reported in the literature [5,22,23,24,27]. Data on patients who were lost to follow-up were gathered to assess whether the loss to follow-up had a significant effect on the observed CHD incidence. Among the 11 patients lost to follow-up for whom follow-up TTE data were not available for assessment, two demonstrated high Cum-5-HIAA levels at death (34 and 35 ULN years). As the ROC analysis indicated a sensitivity of 100% and a specificity of 95% for CHD at 34.7 ULN years, it is plausible that the true incidence of CHD in this population was somewhat higher than observed. If both of these patients with substantial S-5-HIAA exposure had CHD, then the incidence of CHD in the previously undiagnosed population would be approximately 6% during the median follow-up of five years, which is still a significantly lower figure than previously reported.

The performance of proBNP and CgA was inferior to that of Cum-5-HIAA in detecting CHD. The sensitivity of proBNP was just 60%, with a cutoff of 260 ng/L, which is the threshold recommended in the ENETS guidelines [2]. As the prevalence of CHD was limited, the NPV was expectedly high at 95%. The PPV at the same cutoff was 21%. Hence, with this cutoff, proBNP was arguably an inadequate marker for referring the correct individuals to screening for CHD with echocardiography. Our findings suggest that Cum-5-HIAA may demonstrate greater diagnostic performance and that its use may facilitate the appropriate allocation of echocardiography resources. In contrast, vascular function tests did not correlate with echocardiography findings.

Possible reasons for the smaller incidence of CHD found in this study relate to the included patient population, as several previous studies focused exclusively on patients with CS [5,22,23,27]. In this study, 40% of the patients with CHD did not exhibit flushing or diarrhea, the hallmark symptoms of CS, at the time of CHD diagnosis. It is evident that clinicians should be aware of the possibility of CHD development even in the absence of clinical CS. The recent ENETS guidelines recommend assessing patients with elevated urinary 5-HIAA for CHD, regardless of their symptoms [2]. Our data and previous [28] prospective data suggest that assessing the 5-HIAA levels in the blood is a reliable and convenient alternative.

The observed low incidence of CHD may have also been due to the relatively frequent use of treatments that reduced S-5-HIAA levels. Almost all the patients included in this study underwent surgical treatment for the primary tumor and used somatostatin analogs during the follow-up period. PRRT was frequently used, with over half of the patients having had at least one PRRT treatment prior to the time of the follow-up TTE. The treatment of SI-NETs has been positively impacted by the use of PRRT. Since its introduction in 1992 [29], PRRT has undergone considerable development, and the use of ^177^Lu-DOTATATE gained regulatory approval in 2017–2018 [30] after it was shown to improve progression-free survival in a randomized controlled trial [31]. As PRRT has been shown to reduce the serotonin burden in patients with SI-NETs [32], it is likely that its active use in our patient series was partly responsible for the limited incidence of CHD. In addition, CHD was associated with reduced survival, as reported in the literature [5,6,23,27]. Our results support the notion that cumulative serotonin exposure, with cumulative 5-HIAA exposure used as a proxy, is associated with the development of CHD. Further studies are required to determine whether treatments that reduce cumulative 5-HIAA exposure are also able to influence CHD-related mortality.

Aortic PWV was found to be a novel independent risk factor for mortality in SI-NET patients. There are limited data on PWV and malignancies, with one study noting an association of elevated brachial–ankle PWV with a higher incidence of cancer [33], and another observing a link between increased cancer mortality and increased brachial–ankle PWV in patients with type 2 diabetes [34]. To our knowledge, no previous studies have examined the impact that PWV has on survival in populations with a malignant disease. In this study, aortic PWV did not correlate with hepatic tumor load or S-5-HIAA. As no cardiovascular deaths were observed in the population, PWV might function as a general indicator of frailty and elevated mortality. Our findings require confirmation in other patient populations with malignancies before any definite conclusions can be drawn.

The limitations of this study include the loss of some patients to follow-up, as the patients had died. In addition, some of the TTE examinations were not conducted within the initial study setting; the data collected in settings that were different from the initial study setting could be used to diagnose CHD, but this situation reduced the applicability of many CHD scoring systems [15]. Finally, utilizing a larger patient population would have been beneficial, as the incidence of CHD was too low to enable a thorough risk factor assessment for CHD development.

## 5. Conclusions

In conclusion, the findings of this prospective study show that while CHD incidence in patients with SI-NETs is lower than previously anticipated, mortality rates during the median 6 years of follow-up are still substantial, as more than one-third of the patients died. In addition to the presence of CHD, high serum 5-HIAA, and high hepatic tumor load, increased aortic PWV, i.e., high arterial stiffness, was identified as a novel risk factor for mortality in SI-NET patients. Cum-5-HIAA, a proxy of serotonin exposure, emerged as a novel promising biomarker that is simple to calculate for CHD in patients with SI-NETs, while arterial stiffness may represent a novel marker for mortality. The lower incidence of CHD in novel prospective studies of patients with SI-NETs may be explained by better and readily available targeted treatments. However, the mortality rates during 6 years of follow-up in patients with SI-NETs do not seem to directly parallel the observed decrease in CHD. Cum-5-HIAA may be used to improve targeted TTE screening for the early detection of and intervention in patients at increased risk of CHD. Further research is required to validate the findings of this study.

## Figures and Tables

**Figure 1 cancers-16-03896-f001:**
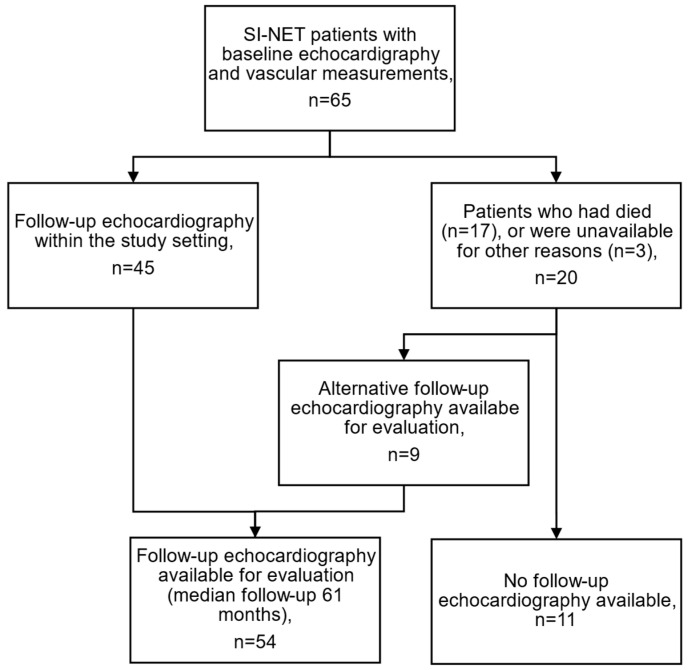
Flowchart of patient inclusion. Of the patients who were unavailable, two did not want to have an additional echocardiography in the study setting, and one had moved abroad and could not be reached. Abbreviations: SI-NET, small intestinal neuroendocrine tumor.

**Figure 2 cancers-16-03896-f002:**
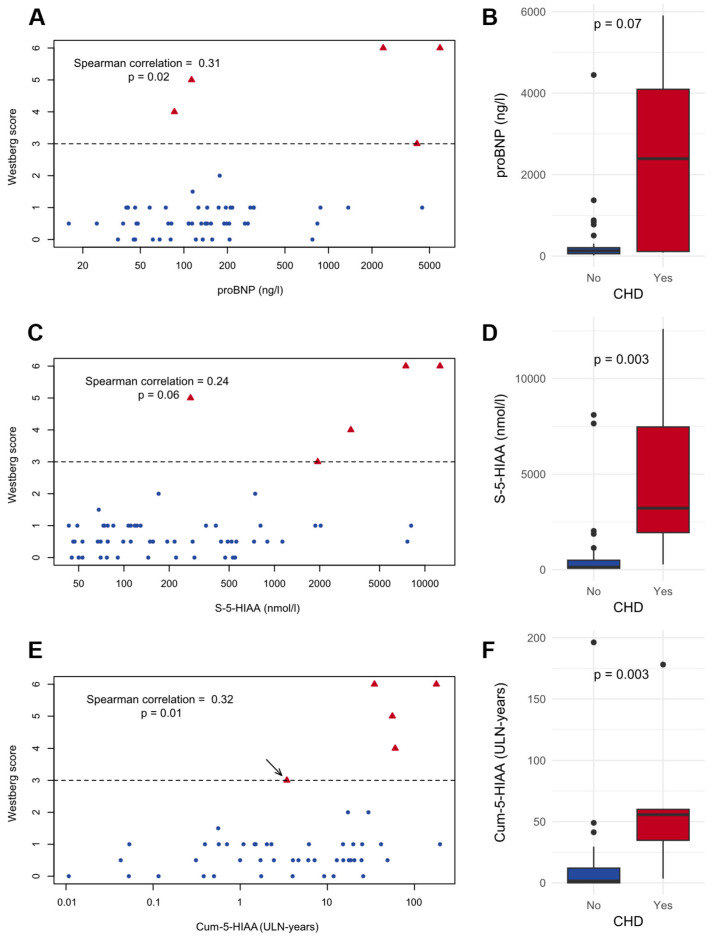
Scatter plots and boxplots comparing the presence of carcinoid heart disease (CHD) and N-terminal pro-brain natriuretic peptide (proBNP) concentration (**A**,**B**); serum 5-hydroxyindoleacetic acid (S-5-HIAA) concentration (**C**,**D**); and cumulative exposure to serum 5-HIAA exceeding the upper limit of normal (ULN) (Cum-5-HIAA) (**E**,**F**). Patients with CHD are shown with red triangles and without CHD with blue circles. The dashed lines denote the Westberg score that was considered diagnostic for CHD. One patient was diagnosed with CHD at the initial diagnostic work-up for small intestinal neuroendocrine tumor, and the measured 5-HIAA exposure was considered unrepresentative. This patient is marked with an arrow.

**Figure 3 cancers-16-03896-f003:**
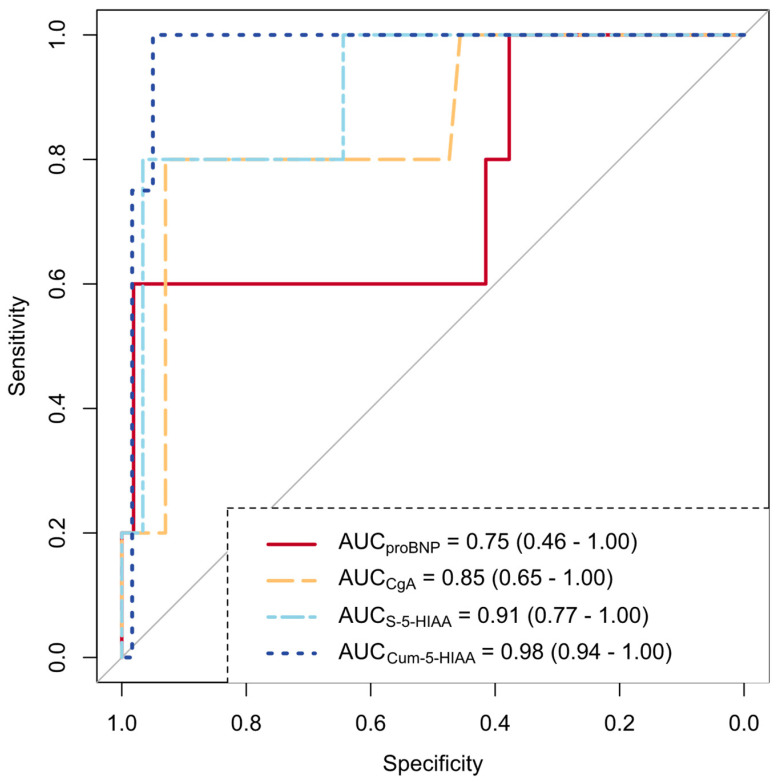
A receiver operating characteristic analysis of 65 patients was conducted to evaluate the performance of N-terminal pro-brain natriuretic peptide (proBNP), chromogranin A (CgA), serum 5-hydroxyindoleacetic acid (S-5-HIAA), and cumulative exposure to serum 5-HIAA exceeding the upper limit of normal (ULN) (Cum-5-HIAA) in the detection of carcinoid heart disease. The area under the curve (AUC) values are shown with the 95% confidence intervals in parentheses.

**Figure 4 cancers-16-03896-f004:**
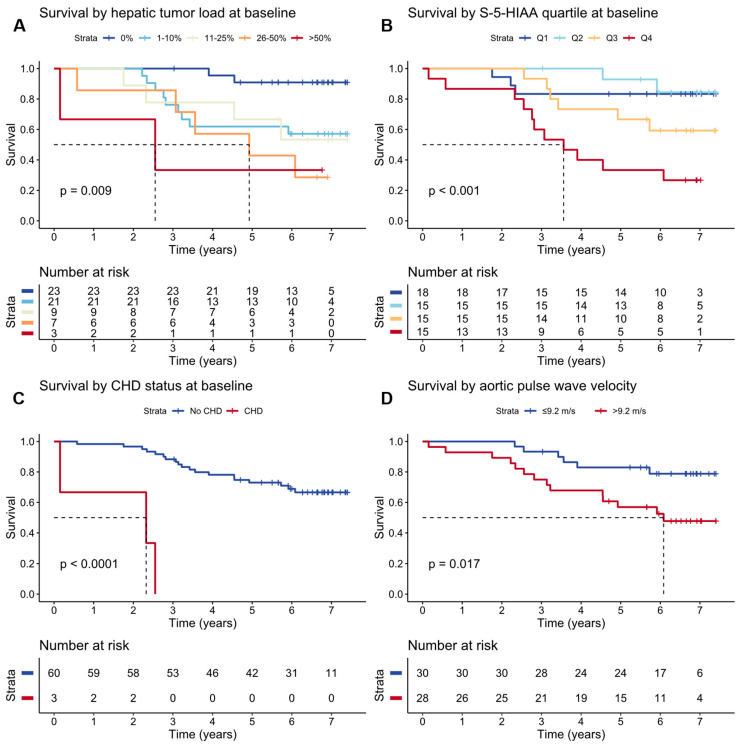
Kaplan–Meier survival estimates for (**A**) initial hepatic tumor load; (**B**) serum 5-hydroxyindoleacetic acid (S-5-HIAA) quartiles; (**C**) carcinoid heart disease (CHD) status; and (**D**) aortic pulse wave velocity. The vertical lines on the survival curves indicate censoring. The dashed lines indicate the time point at which 50% survival was reached. The listed *p*-values are for the log-rank test.

**Table 1 cancers-16-03896-t001:** Patient characteristics.

Variable	All Patients (*n* = 65) at Baseline	Patients Who Underwent Follow-Up TTE (*n* = 54) at Baseline	Patients Who Underwent Follow-Up TTE (*n* = 54) at Follow-Up TTE	Patients Who Did Not Undergo Follow-Up TTE (*n* = 11) at Baseline	CHD Patients (*n* = 5) at CHD Diagnosis
Age (years)	66 (59–72)	64 (58–70)	70 (61–74)	69 (66–75)	62 (57–67)
Sex, female/male (*n*)	33:32 (51%:49%)	27:27 (50%:50%)	27:27 (50%/50%)	6:5 (55%:45%)	1:4 (20%:80%)
Time from the initial SI-NET diagnosis at assessment (months)	72 (32–108)	70 (31–107)	130 (79–169)	87 (38–132)	32 (32–78)
Primary tumor Ki-67 (%)	2 (1–5)	2 (1–5)	2 (1–5)	2 (2–7)	2 (1–3)
Hepatic tumor burden (*n*)					
0%	23 (35%)	20 (37%)	13 (24%)	3 (27%)	0 (0%)
1–10%	23 (35%)	18 (33%)	24 (44%)	5 (45%)	0 (0%)
10–25%	9 (14%)	8 (15%)	9 (17%)	1 (9%)	2 (40%)
26–50%	7 (11%)	6 (11%)	6 (11%)	1 (9%)	1 (20%)
>50%	3 (5%)	2 (4%)	2 (4%)	1 (9%)	2 40%)
Serum 5-HIAA (nmol/L)	138 (78–424)	135 (78–372)	147 (74–533)	286 (78–525)	3220 (1940–7470)
Cum-5-HIAA (ULN years)	0.8 (0.0–4.8)	0.7 (0.0–4.3)	1.9 (0.0–15)	1.0 (0.1–11)	57 (35–60)
Plasma proBNP (ng/L)	81 (35–194)	73 (35–176)	135 (65–237)	128 (56–214)	1283 (113–2391)
CgA (proportion of ULN)	1.7 (0.9–5.5)	1.6 (0.9–5.1)	1.1 (0.6–8.9)	2.3 (1.0–9.0)	83 (32–133)
Treatment (*n*)					
Resection of the primary tumor	57 (87%)	47 (87%)	49 (91%)	10 (91%)	2 (40%)
Resection of recurrence	3 (5%)	2 (4%)	2 (4%)	1 (9%)	0 (0%)
Non-systemic treatment for metastases ^1^	23 (35%)	20 (37%)	25 (46%)	3 (27%)	2 (40%)
Somatostatin analog	56 (86%)	47 (87%)	49 (91%)	9 (82%)	5 (100%)
PRRT ^2^	18 (28%)	15 (28%)	30 (56%)	3 (27%)	3 (60%)
PRRT, retreatment ^2^	3 (5%)	2 (4%)	14 (26%)	1 (9%)	0 (0%)
PRRT, second retreatment ^2^	0 (0%)	0 (0%)	6 (11%)	0 (0%)	0 (0%)
Telotristat ethyl	0 (0%)	0 (0%)	3 (6%)	0 (0%)	0 (0%)
Interferon alfa-2b	12 (18%)	9 (17%)	10 (19%)	3 (27%)	0 (0%)
Chemotherapy ^3^	3 (5%)	2 (4%)	11 (20%)	1 (9%)	0 (0%)

Data are presented as median (interquartile range) or frequency (proportion), as appropriate. Abbreviations: 5-HIAA, 5-hydroxyindoleacetic acid; CgA, chromogranin A; CHD, carcinoid heart disease; Cum-5-HIAA, cumulative ULN-exceeding serum 5-HIAA exposure; proBNP, N-terminal pro-brain natriuretic peptide; PRRT, peptide receptor radionuclide therapy; SI-NET, small intestinal neuroendocrine tumor; TTE, transthoracic echocardiography; ULN, upper limit of normal. ^1^ Including surgical treatment, thermoablation, brachytherapy, and external radiation therapy (excluding palliative treatment). ^2^ Complete treatment finished prior to assessment. ^3^ Chemotherapy agents included everolimus, temozolomide, and combinations of temozolomide and capecitabine.

**Table 2 cancers-16-03896-t002:** Features of the right side of the heart at baseline and follow-up in transthoracic echocardiography (TTE).

Variable	Baseline TTE, All Patients (*n* = 63)		Baseline TTE for Patients with Follow-Up TTE (*n* = 54)		Follow-Up TTE (*n* = 54)	
	Result	Data Available	Result	Data Available	Result	Data Available
Tricuspid valve thickening (*n*)		62/63 (98%)		51/54 (94%)		48/54 (89%)
None	58 (94%)		49 (96%)		46 (96%)	
Mild	1 (2%)		0 (0%)		0 (0%)	
Moderate	3 (5%)		2 (4%)		1 (2%)	
Severe	0 (0%)		0 (0%)		1 (2%)	
Tricuspid valve mobility (*n*)		62/63 (98%)		51/54 (94%)		48/54 (89%)
Increased	0 (0%)		0 (0%)		1 (2%)	
Normal	58 (94%)		48 (94%)		45 (94%)	
Mildly reduced	2 (3%)		2 (4%)		0 (0%)	
Moderately reduced	1 (2%)		1 (2%)		1 (2%)	
Severely reduced	1 (2%)		0 (0%)		1 (2%)	
Tricuspid valve regurgitation (*n*)		61/63 (97%)		50/54 (93%)		50/54 (93%)
None	9 (15%)		7 (14%)		10 (20%)	
Trace	23 (38%)		16 (32%)		17 (34%)	
Mild	24 (39%)		23 (46%)		17 (34%)	
Moderate	3 (5%)		3 (6%)		2 (4%)	
Severe	2 (3%)		1 (2%)		4 (8%)	
Pulmonic valve thickening (*n*)		60/63 (95%)		51/54 (94%)		47/54 (87%)
None	55 (92%)		47 (92%)		45 (96%)	
Mild	3 (5%)		3 (6%)		2 (4%)	
Moderate	2 (3%)		1 (2%)		0 (0%)	
Severe	0 (0%)		0 (0%)		0 (0%)	
Pulmonic valve mobility (*n*)		59/63 (94%)		50/54 (93%)		47/54 (87%)
Increased	0 (0%)		0 (0%)		0 (0%)	
Normal	56 (95%)		48 (96%)		46 (98%)	
Mildly reduced	2 (3%)		1 (2%)		0 (0%)	
Moderately reduced	0 (0%)		0 (0%)		1 (2%)	
Severely reduced	1 (2%)		1 (2%)		0 (0%)	
Pulmonic valve stenosis (*n*)		59/63 (94%)		51/54 (94%)		47/54 (87%)
None	57 (97%)		49 (96%)		46 (98%)	
Mild	1 (2%)		1 (2%)		1 (2%)	
Moderate	1 (2%)		1 (2%)		0 (0%)	
Severe	0 (0%)		0 (0%)		0 (0%)	
Pulmonic valve regurgitation (*n*)		61/63 (97%)		51/54 (94%)		48/54 (89%)
None	39 (64%)		33 (65%)		27 (56%)	
Trace	6 (10%)		5 (10%)		9 (19%)	
Mild	13 (21%)		11 (22%)		11 (23%)	
Moderate	1 (2%)		1 (2%)		0 (0%)	
Severe	2 (3%)		1 (2%)		1 (2%)	
Right ventricle area, systolic (cm^2^) ^1^	12 (8–16)	56/63 (89%)	12 (9–16)	47/54 (87%)	11 (8–17)	47/54 (87%)
Right ventricle basal dimension, diastolic (mm) ^1^	34 (31–40)	59/63 (94%)	34 (31–40)	49/54 (91%)	35 (30–39)	44/54 (81%)
Right ventricle mid-cavity dimension, diastolic (mm) ^1^	31 (27–36)	58/63 (92%)	31 (28–35)	49/54 (91%)	32 (29–36)	47/54 (87%)
Right ventricle longitudinal dimension, diastolic (mm) ^1^	63 (59–67)	58/63 (92%)	63 (60–67)	49/54 (91%)	65 (60–69)	49/54 (91%)
Right atrium area, systolic (cm^2^) ^1^	13 (15–19)	58/63 (92%)	15 (13–18)	48/54 (89%)	15 (14–19)	44/54 (81%)
Tricuspid annular plane systolic excursion, TAPSE (mm)	20 (20–25)	57/63 (90%)	22 (21–25)	47/54 (87%)	22 (20–24)	50/54 (93%)
Westberg score	1 (0.5–1) [0–6]	61/63 (97%)	1 (0.5–1) [0–4]	50/54 (93%)	0.5 (0.5–1) [0–6]	47/54 (87%)

Data are presented as median (interquartile range)/[range] or frequency (proportion), as appropriate. ^1^ Evaluated from apical four-chamber view.

**Table 3 cancers-16-03896-t003:** Baseline characteristics by vital status at the end of the follow-up.

Variable	Alive (*n* = 42)	Deceased (*n* = 22)	*p*-Value
Age (years)	65 (60–70)	66 (59–74)	0.31
Sex, female/male (*n*)	24/18 (57%/43%)	8/14 (36%/64%)	0.19
Time from the initial SI-NET diagnosis at assessment (months)	75 (47–109)	77 (32–108)	0.85
Primary tumor Ki-67 (%)	2 (1–5)	2 (2–5)	0.24
Hepatic tumor burden (*n*)			0.006
0%	20 (47%)	2 (9%)	
1–10%	14 (33%)	9 (41%)	
10–25%	5 (12%)	4 (18%)	
26–50%	2 (5%)	5 (23%)	
>50%	1 (2%)	2 (9%)	
Serum 5-HIAA (nmol/L)	95 (70–183)	433 (174–746)	<0.001
Cum-5-HIAA (ULN years)	0.3 (0.0–1.2)	6.3 (0.7–14)	<0.001
Plasma proBNP (ng/L)	55 (35–176)	109 (57–214)	0.10
CgA (proportion of ULN)	1 (1–3)	6 (2–14)	<0.001
Treatment (*n*)			
Resection of the primary tumor	39 (93%)	17 (77%)	0.11
Resection of recurrence	2 (5%)	1 (5%)	1.0
Non-systemic treatment for metastases ^1^	17 (40%)	6 (27%)	0.41
Somatostatin analog	35 (83%)	21 (95%)	0.25
PRRT ^2^	10 (24%)	8 (36%)	0.38
PRRT, retreatment ^2^	1 (2%)	2 (9%)	0.27
PRRT, second retreatment ^2^	0 (0%)	0 (0%)	n/a
Telotristat ethyl	0 (0%)	0 (0%)	n/a
Interferon alfa-2b	4 (10%)	8 (36%)	0.04
Chemotherapy ^3^	1 (2%)	2 (9%)	0.27

Data are presented as median (interquartile range) or frequency (proportion), as appropriate. Abbreviations: 5-HIAA, 5-hydroxyindoleacetic acid; CgA, chromogranin A; CHD, carcinoid heart disease; Cum-5-HIAA, cumulative ULN-exceeding serum 5-HIAA exposure; proBNP, N-terminal pro-brain natriuretic peptide; PRRT, peptide receptor radionuclide therapy; SI-NET, small intestinal neuroendocrine tumor; TTE, transthoracic echocardiography; ULN, upper limit of normal. ^1^ Including surgical treatment, thermoablation, brachytherapy, and external radiation therapy (excluding palliative treatment). ^2^ Complete treatment finished prior to assessment. ^3^ Chemotherapy agents included everolimus, temozolomide, and combinations of temozolomide and capecitabine.

**Table 4 cancers-16-03896-t004:** Determinants of overall survival in univariate and multivariate analysis using Cox proportional hazards model.

Variable	Univariate Hazard Ratio (95% CI)	*p*-Value	Multivariate Hazard Ratio (95% CI)	*p*-Value
Sex				
Female	1		1	
Male	1.88 (0.79–4.47)	0.16	1.29 (0.46–3.61)	0.62
Age at TTE	1.03 (0.97–1.09)	0.20	1.05 (0.98–1.13)	0.20
Serum 5-HIAA (nmol/L)	1.00 (1.00–1.00)	0.01	1.00 (1.00–1.00)	0.60
Aortic pulse wave velocity (m/s)	1.23 (1.09–1.40)	0.001	1.22 (1.04–1.43)	0.01
Metastases at baseline				
No	1		1	
Yes	7.02 (1.68–30.8)	0.008	5.28 (1.09–25.6)	0.04
CHD at baseline				
No	1		1	
Yes	24.8 (5.43–113.4)	<0.001	36.1 (5.36–243)	<0.001

Abbreviations: 5-HIAA, 5-hydroxyindoleacetic acid; CHD, carcinoid heart disease; CI, confidence interval.

## Data Availability

The data used in this study are not publicly available as the study participants have not given consent to data sharing.

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
