# Peer review of "The Value of Repeat 5-HIAA Measurements as a Predictor of Carcinoid Heart Disease: A Prospective 5-Year Follow-Up Study in Patients with Small Intestinal Neuroendocrine Tumors"

_cancers, 2024, doi:10.3390/cancers16233896_

Round 1
Reviewer 1 Report
Comments and Suggestions for Authors
This is a manuscript by Kostiainen I et al presenting the Cum-5HIAA as a new biomarker for CHD. The manuscript is well written.
My main argument is how easy it is going to use the Cum formula in daily clinical praxis.
The authors have also named that the cases are very few. More patients with CHD would strengthen the conclusion drawn.
Author Response
We thank the reviewer for their commentary.
Comments 1: My main argument is how easy it is going to use the Cum formula in daily clinical praxis.
Response 1: Calculation of cumulative 5-HIAA exposure is indeed going to require some additional effort. Fortunately, the calculation of cumulative exposure is simple, and could be facilitated by a electronic form that requires just results and dates of 5-HIAA measurements.
Comment 2: The authors have also named that the cases are very few. More patients with CHD would strengthen the conclusion drawn
Response 2: We agree with the reviewer. Validation of our results necessitates other prospective studies.
Reviewer 2 Report
Comments and Suggestions for Authors
It is a very interesting and well-written paper that fits the scope of the journal.
The introduction successfully describes the current state of the research field, highlights the limitations that currently exist in this area, and mentions the study's objective and why it is relevant.
The study is well-designed and the material and methods section is easy to understand.
The results section is easy to read and the results are well presented.
Despite being already published, I would like to authors to clarify what the hepatic tumor burden is and what the different percentages mean.
In the discussion section, the authors do a good evaluation of their results and critically discuss them.
Author Response
We thank the reviewer for their commentary.
Comment 1: Despite being already published, I would like to authors to clarify what the hepatic tumor burden is and what the different percentages mean.
Response 1: We have now provided further explanation of the calculation of hepatic tumor burden in the methods section.
Reviewer 3 Report
Comments and Suggestions for Authors
The manuscript highlights the significance of Cum-5-HIAA as a promising predictive biomarker for CHD in SI-NET patients. The prospective, five-year follow-up design offers a comprehensive view of the clinical impacts of cumulative serotonin exposure, focusing on CHD and survival rates. This manuscript provides a well-conducted analysis with significant clinical implications.
Here are some suggestions for improving the manuscript:
1 Please verify the correct use of punctuation in the methods section of the abstract.
2 Please organize the Methods and Results sections into clear, logical subsections that align with your main study aims.
3 Please adjust the table widths for better presentation.
4 Consider adding a comparison table for biomarker diagnostic performance.
5 Conclude the results section with a brief and concise summary of major findings.
Author Response
We thank the reviewer for their commentary.
Comment 1: Please verify the correct use of punctuation in the methods section of the abstract.
Response 1: We thank reviewer for the suggestion. The manuscript underwent additional language editing to improve clarity.
Comment 2: Please organize the Methods and Results sections into clear, logical subsections that align with your main study aims.
Response 2: We have added further subsections in the Methods section to improve the clarity of the manuscript.
Comment 3: Please adjust the table widths for better presentation.
Response 3: We have made some adjustments for table width to improve the reading experience.
Comment 4: Consider adding a comparison table for biomarker diagnostic performance.
Response 4: We thank the reviewer for this suggestion. Comparison of biomarker performance is provided in ROC analysis (Figure 3). While additional table could provide some clarity, it would be somewhat redundant in this context. While we agree that comparison table with, for example, sensitivity and specificity data for given biomarker values could be informative. Providing such a table would necessitate subjective selection the cut-offs. As such, we prefer to use and present ROC analysis as an objective method to display diagnostic performance.
Comment 5: Conclude the results section with a brief and concise summary of major findings.
Response 5: We have now provided a conclusion section as per request.